# Experimental Study of the Shear Performance of Combined Concrete–ECC Beams without Web Reinforcement

**DOI:** 10.3390/ma16165706

**Published:** 2023-08-20

**Authors:** Kai Cheng, Yulin Du, Haiyan Wang, Rui Liu, Yu Sun, Zhichao Lu, Lingkun Chen

**Affiliations:** 1Fourth Engineering Company Limited, China National Chemical Communications Construction Group Co., Ltd., Binzhou 256200, China; chengkai@zhxjj.com.cn (K.C.); liur@zhxjj.com.cn (R.L.); 2Shijiazhuang Institute of Railway Technology, Shijiazhuang 050041, China; 3College of Transportation Science & Engineering, Nanjing Tech University, Nanjing 211816, China; why8621@163.com (H.W.); sunyu.9003@163.com (Y.S.); 4College of Architecture Science and Engineering, Yangzhou University, Yangzhou 225127, China; mx120220593@stu.yzu.edu.cn

**Keywords:** concrete–ECC composite beam, shear performance, parametric analysis, FE simulation

## Abstract

Background: Shear damage of beams is typically brittle damage that is significantly more detrimental than flexural damage. Purpose: Based on the super-high toughness and good crack control ability of engineered cementitious composites (ECC), the shear performance of concrete–ECC beams was investigated by replacing a portion of the concrete in the tensile zone of reinforced concrete beams with ECC and employing high-strength reinforcing bars to design concrete–ECC beams. The purpose of this investigation is to elucidate and clarify the shear performance of concrete–ECC beams. Methodology/approach: Experimental and FE analyses were conducted on the shear performance of 36 webless reinforced concrete–ECC composite beams with varied concrete strengths, shear-to-span ratios, ECC thicknesses, and interfacial treatments between the layers. Results: The results indicate that the effect of the shear-to-span ratio is greater, the effect of the form of interface treatment is smaller, the effect is weakened after the ECC thickness is greater than 70 mm (i.e., the ratio of the replacement height to section height is approximately 0.35), the shear resistance is reduced when the hoop rate is greater, and the best shear resistance is obtained when the ECC 70 mm thickness and the hoop rate of 0.29% are used together. Conclusions: This study can serve as a technical reference for enhancing the problems of low durability and inadequate fracture control performance of RC beams in shear and as a guide for structural design research.

## 1. Introduction

Reinforced concrete materials are the most widely used construction materials due to their advantages in mechanical properties and engineering costs [1]. However, concrete materials are prone to cracking due to their characteristics. If cracks are not properly controlled, they will gradually expand under external loads, leading to the penetration of harmful substances into the interior, peeling off of the protective layer, corrosion of the reinforcement, and other problems, which will seriously reduce the structure’s durability [2]. Engineered cementitious composites (ECCs) are a new type of material formed by adding short fibers with appropriate specific properties in a disordered distribution to the cement matrix [3]. The concrete material is similar to metallic materials, which can improve material brittleness, toughness, and durability [4].

Shear resistance design has been a standard topic in structural design, and although the ductility characteristics of concrete can be increased by setting hoop reinforcements, the effect of transmitting the shear force by the hoop and longitudinal reinforcement alone is less than ideal [5]. Hippola et al. [6] proposed a novel FE cell, which was then subjected to a comprehensive validation process, including 170 tests to assess its correctness. The conducted tests exhibited variations in many key parameters, including shear-span-to-depth ratios, rates of longitudinal and transverse reinforcement, concrete strength, section depth, boundary conditions, and distinct mechanisms of damage. A novel FE cell was developed with the aim of conducting a full investigation of the shear mechanism in reinforced concrete. Recent research has shown that the enhancement of existing reinforced concrete structures is not feasible without external interventions. Therefore, it is necessary to avoid this shear damage through good resistance design. In contrast, the matrix material of reinforced ECC (RECC) can be coordinated with the deformation of the reinforcement, so the ECC shows fine steady-state characteristics of diagonal cracks when subjected to shear, and the shear damage has similar ductility characteristics, which can be used in structures to achieve the shear resistance requirements.

Regarding ECC’s unique properties, Li et al. [7] designed ordinary reinforced cement (RC) beams with a 0.75% hoop ratio, fiber-reinforced cement (FRC) beams with different fiber references, ECC mixed with 7% by volume Dramix fibers (DRECC) beams, and spectra fiber ECC (SPECC) beams for shear comparison tests. The test results showed that the DRECC beams with high tensile strength and moderate tensile strain were damaged at a shear stress of 9.89 MPa, 300% higher than the plain concrete beams and 81% higher than the RC beams, while the shear strength of the SPECC beams without webs was comparable to that of the RC beams with a 0.75% hoop ratio, and the shear performance of the ECC beams was significantly better than that of the FRC and RC beams. The findings from the ECC beam shear experiments conducted by Kanda et al. [8] demonstrated that, the shear compression and shear tension damage occurred in the ECC beams under the action of cyclic cycles, and the member bearing capacity was increased by 50% compared with that of ordinary concrete beams, where the ultimate deflection in shear tension damage was increased by two times showing the ductility characteristics. Fukuyama et al. [9] studied the ability of ECC to reduce the degree of seismic response and shear damage with the polyvinyl alcohol-engineered cementitious composite (PVA-ECC) beam cyclic load test. The results showed that PVA-ECC could improve the members’ structural shear performance and damage tolerance. Shimizuet et al. [10] designed ECC beam shear tests with the fiber admixture and hoop rate as variables and a shear-to-span ratio of 1.5. Zij et al. [11] studied the shear performance of SHCC beams with fiber doping as a variable; the results showed that the cracks were sparse at less than 2% fiber doping in pure shear stress conditions, and dense cracks were produced at the notch at more than 2% with the highest shear strength and better reliability of shear resistance. Park et al. [12] designed shear tests of strain-hardening, fiber-reinforced cement-based composite (SHCC) beams with steel fibers, polyethylene fibers, and prebuilt materials, and found that all three materials significantly improved the shear strength of RC beams compared with ordinary concrete members. Alyousif [13] designed shear tests of beams without webs using the shear-to-span ratio and reinforcement ratio as parameters, and the results showed that Ryerson mix concrete (RMC) (i.e., ultra-high-strength fiber-reinforced cementitious composite) beams have a higher shear-bearing capacity and yield stiffness than ECC beams. In contrast, ECC beams have a higher deflection ductility ratio and energy absorption capacity, and both types of materials significantly limit shear cracking. Hung et al. [14] designed U-shaped sheathing with ECC materials to reinforce the shear defect areas of RC cantilever beams, where the contact interfaces were untreated steel reinforcement and wire mesh. The results show that all three forms can significantly improve the strength, stiffness, ductility, energy dissipation capacity, and shear deformation of the original RC members, but the untreated interface can simplify the construction process.

In recent years, Hou et al. [15] designed six shear tests of ECC beams without web reinforcement using the reinforcement ratio ρ and shear-to-span ratio λ as variables. The results showed that the shear strength of ECC beams at λ = 2.04 and ρ = 2.28% was 14.3% higher than that of the RC control beams, while the shear strength of ECC beams at ρ = 4.25% was 8.3% higher than that at 2.28%, indicating that the shear resistance of ECC was more significant than that of longitudinal reinforcement. Yang et al. [16] studied the effects of reinforcement rate, shear-to-span ratio, and hoop rate on the shear resistance of ECC beams. From the test, it was found that the ultimate load of the ECC beams with web reinforcement was 25.5% higher than that of the RC control beams, and the effect of each variable on the cracking load of the ECC beams was small. The effect of the shear-to-span ratio on shear force is greater than that of the reinforcement ratio, and the reduction in the shear force from 1 to 3 is nearly 45%. Ji et al. [17] designed shear experiments of ECC beams with large shear-to-span ratios, hoop ratios, and fiber doping as parameters. The results showed that the cracking load was almost independent of any factors, and the variables other than fiber admixture had no significant effect on the crack width, but the fiber admixture had a slight effect on the crack width when it exceeded 2%. Wang et al. [18] concluded from the shear test of ECC short beams that ECC can improve the members’ shear-bearing capacity and shear ductility. The shear-bearing capacity was calculated by using the tensile compression bar model, FE method, and our code, and the values obtained with the tensile compression bar and FE method were in good agreement with the measured values.

In contrast, the values calculated with the code method were small, and the differences in the test values were more than half. Deng et al. [19] investigated ECC’s shear performance and deformation level and steel ECC’s combined short beams with the shear-to-span ratio and reinforcement ratio as variables. The test results showed that the shear-to-span ratio and reinforcement ratio greatly affect the shear-bearing capacity and shear damage pattern, and the shear force decreases with increasing shear-to-span ratio and increases with increasing reinforcement ratio.

In order to study the shear performance of ECC in composite beams and to solve the cracking problem of composite beams, in this paper, based on the existing research on ECC performance, the shear performance of concrete–ECC composite beams is investigated using the design of PVA fibers. The effects of concrete strength, shear-to-span ratio, ECC thickness, and interface treatment on the shear performance of concrete–ECC composite beams are analyzed, and a FE model is established for verification based on experiments. The results can be used as a reference for the design of concrete–ECC composite beams.

## 2. Experimental Overview

### 2.1. Component Design

In this test, 36 specimens of supported beams without web reinforcement were designed, and the geometry and construction of the specimens are shown in Figure 1. The design parameters are shown in Table 1, and the rules for naming the specimens are as follows:

(1)“B” means beam without web;(2)“J0” denotes RC (reinforced concrete) control beam;(3)“J1” indicates the test beam with wire mesh at the concrete–ECC interlayer interface;(4)“J2” indicates the test beam with a hemispherical concave surface at the concrete–ECC interlayer interface. Figure 2 provides a clear depiction of the composition and arrangement of the wire mesh and hemispherical concave surfaces at the interface between the concrete and ECC sandwich. It illustrates the specific constituents and their respective configurations inside the beam.(5)“C” indicates concrete; the number after the character “C” indicates the standard compressive strength of concrete;(6)The number after “E” indicates the thickness of the ECC layer.

In order to better describe the experimental procedure, Table 1 is divided into three different tables; Table 1-1 includes the beams with 70 ECC thickness (J1 and J2) and the control beam (J0); Table 1-2 includes the beams with 100 ECC thickness; Table 1-3 includes the C30 strength beams and the beams with 100 ECC thickness.

### 2.2. Material properties

The materials used to make the components are mainly ordinary concrete, steel reinforcement, ECC, and fine wire mesh. Among them, the materials for concrete are cement, sand, stone, water, and water reducer; the materials for making ECC are mainly fly ash, cement, silica fume, quartz sand, PVA fiber, water, and water reducer. Some of the tested raw materials are shown in Figure 3.

#### 2.2.1. Cement

The cement used for the preparation of the concrete and ECC materials in this test was P.O42.5 ordinary silicate cement produced by Yangzhou Green Yang Cement Company (Yangzhou, China), and the main physical properties of this cement are shown in Table 2.

#### 2.2.2. Fly Ash and Silica Fume

The fly ash used in this paper is the Ⅰ grade ash of Nanjing Thermal Power Plant (Nanjing, China), whose performance is shown in Table 3; the silica fume is produced from Huzhou, Zhejiang Province, and its performance is shown in Table 4.

#### 2.2.3. Quartz Sand

The quartz sand used for the ECC in this paper comes from the Anhui Fengyang Shengli Sand Factory. The particle size of the quartz sand is in the range of 100~200 mesh. Quartz sand is sand with high purity, high acid and alkali resistance, high refractoriness, and a wide range of uses. Table 5 shows the performance index of this quartz sand.

#### 2.2.4. Fibers

This test uses the short-cut 12-type PVA fiber produced by Fujian Baohua Company (Quanzhou, China), and Table 6 shows the performance index of the PVA fiber.

#### 2.2.5. Water and Water-Reducing Agents

The water used in this test is Yangzhou municipal tap water. The water-reducing agent used is the standard polycarboxylic acid water-reducing agent produced by Suzhou Sika Company (Suzhou, China), with a water reduction rate of 20%~40%; the quality is in accordance with the requirements of Concrete Admixtures GB8076-2008 [20].

#### 2.2.6. Normal Concrete Aggregate

Considering the size of the components and the spacing of the reinforcement skeleton, the fine aggregate used in the production of concrete is the natural river sand in Yangzhou, intermediate; the coarse aggregate is the crushed stone with a better set of grain sizes of 5~20 mm. The materials are produced by a local sand and gravel plant in Yangzhou, and the main technical parameters are shown in Table 7 and Table 8.

Two groups (three per group) of C30 concrete, C50 concrete, and ECC cubic specimens were cast, respectively, with concrete specimen sizes of 150 mm × 150 mm × 150 mm and an ECC specimen size of 70.7 mm × 77.7 mm × 77.7 mm. Cubic compressive strength tests were conducted, which were synchronized with the beam tests, and the C30 concrete, C50 concrete, and the measured cubic compressive strengths of the ECC were 35.33, 54.45, and 52.85 MPa, respectively. The mix ratios of the ECC and concrete are shown in Table 9 [21].

### 2.3. Measurement Content and Program

The test was loaded symmetrically at four points, as shown in Figure 4. Fixed and sliding hinge supports were set at both ends of the test beam. In order to prevent local pressure damage at the test piece’s support point and the loading point’s location, a steel mat was placed at each direct stress point.

Load, strain, displacement, and crack width were measured in the test. The load was displayed and collected in real time by a strain gauge connected to the sensor; the displacement was measured and collected by three percentage meters. Based on the properties of shear damage, it is necessary to establish the placement location of the concrete strain gage paste. Specifically, the strain gages should be arranged equidistantly along the line connecting the loading point and the support point, two strain gages should be positioned at the mid-span of the beam. The loading point is represented by arrows, while the strain gages are denoted by numbers 1 to 8, as illustrated in Figure 5. The crack measurement was carried out by visual observation; a crack observer measured the crack width, and the larger value was recorded as the representative value of cracks at this level.

## 3. Test Results and Analysis

### 3.1. Experimental Phenomena

When the shear span is relatively small, both the RC beam and the concrete–ECC combination beam are damaged by diagonal compression. However, the brittle damage of the RC beam is obvious, such as in beam B1. During the loading process, the surface of the shear span area appears with concrete broken off, and finally, the member is fractured and has obvious dislocation. In beam B15, fewer cracks were produced during the process. Only a major diagonal crack was produced in the area, and there was no warning when the damage occurred. While the concrete at the loading point of the combined beam was damaged, the phenomenon was not obvious, as in beam B15 the phenomenon of surface material shedding did not occur in the shear span area, and it still maintained good integrity. The number of cracks in the shear compression area was much more than in the RC beam, and there are many relatively small diagonal cracks near the location of the main diagonal crack in the limit state that can be seen in the ECC area between the cracks, with fiber bonded to the fractured part. The damage patterns of beams B1 and B15 are shown in Figure 6.

When the shear-to-span ratio is moderate, shear damage occurs in the test beam. As in the case of RC beam B18, vertical cracks first appear at the lower edge of the concrete in the tension zone, and the cracks can be seen to extend and widen rapidly once the concrete in the tension zone is cracked. Eventually, the cracks in the diagonal direction run through the cross-section of the RC beam, the concrete in the compression zone between the two loading points is severely broken locally, the crack width in the tension zone is large, and the longitudinal reinforcement is subjected to large yielding or near yielding. While the vertical cracks of the concrete–ECC combination beam do not expand rapidly to the support point after appearing, the number of cracks increases continuously, which limits the expansion of the main cracks well. When the load reaches a certain level, the secondary diagonal cracks in the concrete part grow to form the main diagonal cracks. The diagonal cracks do not penetrate the whole cross-section like the RC beam, but occur in the ECC part to spread into multiple diagonal cracks with smaller widths. The concrete near the loading point is crushed under the action of shear pressure, in which the crack control performance of the ECC with the 100 mm thickness is slightly better than that of the 70 mm thickness, such as beams B7 and B24. The damage patterns of beams B18, B7, and B24 are shown in Figure 7.

When the shear span is relatively large, the beam suffers diagonal tensile damage. In the case of RC beam B21, the diagonal crack appears rapidly at the loading point, the concrete surface near the loading point crumbles, the width is visible to the naked eye, and the number of bending zones is small, but the cracks are long. A typical diagonal tensile crack through the whole interface is formed in the direction of the line connecting the loading point and the support point, causing the beam to be pulled off and damaged along the diagonal direction. In contrast, the main diagonal crack of the combined beam is not along the diagonal line. However, bulges above the diagonal line, similar to an arch, and the lower end of the main crack do not extend to the bottom of the beam but are roughly evenly dispersed to disappear at the height of the ECC layer stack, and the damage is accompanied by an obvious “bared” fiber tearing sound, as in the case of beam B14. The damage patterns of beams B21 and B14 are shown in Figure 8.

### 3.2. Concrete–ECC Shear-Bearing Capacity Analysis

#### 3.2.1. The Effect of Concrete Strength on Shear-Bearing Capacity

The effect of concrete strength in the compression zone on the shear strength of the concrete–ECC beams is shown in Figure 9, where the vertical coordinate is the shear-bearing capacity V, and the horizontal coordinate is the shear-to-span ratio λ.

It should be noted that Figure 9 illustrates the impact of concrete strength on the shear loads in concrete–ECC beams. This serves as an illustrative example for beams without web reinforcement (E70 thick series) and beams with web reinforcement (J1E70 thick series).

Figure 9 illustrates the impact of concrete strength on the combined beams lacking web reinforcement. Notably, this effect is more prominent in the concrete–ECC beams compared with the RC beams. Specifically, the average increase in shear strength for the beams lacking web reinforcement is 32% when the concrete strength grade is elevated from C30 to C50. In contrast, the average increase in shear strength for the beams with web reinforcement is 10%. Furthermore, it has been shown that the enhancement in the shear strength of beams has a diminishing trend as the shear-to-span ratio increases, particularly when the concrete strength class transitions from C30 to C50. One possible explanation for this phenomenon is that as the shear-to-span ratio increases, the damage type of the beam transitions from inclined compression damage to inclined tensile damage. This transition occurs due to changes in the internal force transmission within the beam. Consequently, the influence of material strength on the shear capacity of the beam also changes [22,23].

For instance, the compressive strength of the concrete significantly affects the shear strength of the beam during inclined compression damage. Conversely, the tensile properties of the material in the tension zone play a crucial role in the occurrence of inclined tensile damage in the beam. In the context of diagonal compression damage in beams, the influence of concrete compressive strength on beam shear strength is observed to be more pronounced. Conversely, in the case of diagonal tension damage in beams, the tensile properties of the material within the tensile zone exhibit an augmented role, leading to a corresponding decrease in the impact of the concrete compressive strength [24,25].

When comparing ECC material to concrete, it is seen that ECC material does not exhibit significant advantages in terms of compressive performance. However, it does possess better tensile qualities, making it appropriate for implementation in the tensile zone of beams. This application serves to enhance the shear brittleness of the beam [26].

To examine the shear behavior of concrete–ECC beams, Figure 10 illustrates the force diagram of concrete–ECC beams subjected to shear. The diagram includes various forces: Vcf, representing the shear force provided by ECC; D_c_, which denotes the compressive stress in the shear-compression zone and is analogous to the compressive properties of the ECC and concrete observed in RC beams; V_af_, indicating the bridging force supplied by fibers at the diagonal cracks; T_sf_, representing the combined force of the tensile reinforcement pulling force after fiber reinforcement; and V_df_, signifying the pinning force exerted by the tensile reinforcement and ECC.

Taking the concrete–ECC beam without web reinforcement as an example, its shear capacity and shear-to-span ratio can be obtained as the dimensionless parameter *V/*(*f*_c_*bh*_0_); the equation of the relationship between *V/*(*f*_c_*bh*_0_) and the shear-to-span ratio is
(1)$$\frac{V}{f_{c}bh_{0}}=k_{1}(\lambda )+k_{2}(\lambda )\eta \sigma_{E}/f_{c}$$

The test data of the webless beams in this work were evaluated using the program Origin to fit the lower envelope of the data. The resulting relationship of *k*_1_(*λ*) is shown in Equation (2). The contrast between the equation represented by Equation (2) and the experimental data is seen in Figure 11; i.e., the value of λ was determined using Equation (2). Figure 11 illustrates the contrast between λ and the experimental results, the scatter dot in Figure 11 is from experimental data. Next, the concrete shear contribution *V*_c_ was deducted from the shear capacity *V* to find the shear contribution *V*_f_ of ECC; this was performed using the test data obtained from the webless reinforced beams. The resulting connection between V_f_ and the shear-to-span ratio is represented by Equation (3). The formula for calculating the shear capacity of a webless reinforced concrete–ECC combination beam may be expressed as Equation (4), or more precisely, Equation (5).
(2)$$k_{1}(\lambda )=0.7/(7\lambda +1)$$
(3)$$V_{f}/(\eta \sigma_{E}bh_{0})=k_{2}(\lambda )$$
(4)$$V=V_{c}+V_{f}$$
(5)$$V=0.7f_{c}bh_{0}/(7\lambda +1)+0.35\sigma_{E}bh_{0}/(1.3\lambda +1)$$

In this study, the impact coefficient of the fiber reinforcing material is denoted as η. Two different thicknesses of ECC, namely, 70 mm and 100 mm, were designed and evaluated. It was shown that the augmentation effect diminishes beyond an ECC thickness of 70 mm. The available scholarly literature [27] pertaining to the beam bending test of a concrete–ECC combination indicates that the increase provided by ECC becomes less significant when the thickness of the ECC exceeds 0.2 H. Based on the findings presented in this study, it is seen that the calculation outcomes exhibit greater levels of satisfaction when $\eta =h_{\mathrm{ECC}}/h$=0.35. Upon further examination, it is evident that the thorough evaluation of ECC thickness and hoop rate demonstrates superior outcomes when using a thickness of 70 mm as opposed to 100 mm. Moreover, the most optimal outcome is achieved when the hoop rate is 0.29%.

#### 3.2.2. Effect of Shear-to-Span Ratio on Shear-Bearing Capacity

The effect of the shear-to-span ratio on the shear-bearing capacity of concrete–ECC beams is shown in Figure 12, whose vertical coordinate is the shear-bearing capacity V and whose horizontal coordinate is the shear-to-span ratio λ. As can be seen from the figure, the shear-to-span ratio is an important factor affecting the shear-bearing capacity of the concrete–ECC beam, and it is observed that the shear-bearing capacity of cracking and ultimate load decreases faster with the increase in the shear-to-span ratio when the shear-to-span ratio is less than 2; after the shear-to-span ratio is greater than 2, the shear-bearing capacity of the beam decreases more slowly with the increase in the shear-to-span ratio. At the same time, the cracking load of the combined beam also decreases with the increase in the shear-to-span ratio, but the drop of the cracking load is less than the ultimate load [28]. When the shear-span quartic ratio of the beam is greater than 3, some of the beams have diagonal tensile damage, but the diagonal cracks are deflected at the ECC interface layer, and eventually, no through cracks are formed, and the loading point of another part of the beam also displays a local crushing phenomenon [29].

#### 3.2.3. Effect of ECC Layer Thickness on Shear-Bearing Capacity

The effect of the ECC layer thickness on the shear-bearing capacity of the combined beam is shown in Figure 13, whose vertical coordinate is the shear-bearing capacity V and whose horizontal coordinate is the shear-to-span ratio λ. Compared with the reinforced concrete beams, the shear-bearing capacity of all concrete–ECC beams, except beam B3, increased to different degrees. Compared with the RC beams, the increase in the shear-bearing capacity in the range of shear-to-span ratio in this paper increases with the increase in the shear-to-span ratio for the beams without web reinforcement at an ECC layer thickness of 70 mm; the mean increase in the ultimate shear-bearing capacity of the beams is 6%, and the mean increase in the cracking load is 36%. At an ECC layer thickness of 100 mm with higher shear-bearing capacity, the shear-to-span ratio increases in the range of 1~3 compared with the reinforced concrete beams under the same conditions. The increase within the shear-to-span ratio increases with the increase in the shear-to-span ratio; the average value of the increase in the shear-bearing capacity is 27%, and the amplitude of the cracking load is 25%. As the concrete in the tension zone is replaced by ECC, it increases the tensile and shear properties of the lower part of the beam [30], which makes up for the defects of the brittle concrete that makes the lower part of the concrete prone to cracking and failure and is beneficial to increasing the overall force performance [31]. However, because the reinforcement also has excellent tensile properties, a higher ECC thickness is not necessarily better [32,33].

The results of the present study show that, in comparison with RC beams subjected to the same conditions, the average increase in shear capacity for beams with a combination of web reinforcement within the designed shear-to-span ratio is 6%. Additionally, the corresponding increase in the cracking load is 26%. It is worth noting that beams with a hoop ratio of 0.29% exhibit a larger increase with the thickness of the ECC layer. Specifically, when the ECC layer thickness is 100 mm, the mean increase is 9%, whereas, for a thickness of 70 mm, the mean increase is 6%. Conversely, beams with a hoop ratio of 0.42% experience only a 2% increase when the ECC layer thickness is 70 mm, resulting in a 6% increase. The experimental results indicate that the increase in the measured value is 6% for an ECC thickness of 70 mm, while the mean increase in the measured value is only 2% for an ECC thickness of 100 mm. These findings indicate that the augmentation of shear in beams with web reinforcement is more pronounced when the thickness of ECC is increased, particularly at lower hoop rates. However, as the hoop rate increases, the impact of increasing ECC thickness on enhancing the shear capacity diminishes, and in some cases, the shear capacity of certain beams slightly decreases. In terms of cracking load, the outcomes for beams with and without web reinforcement exhibit similar trends.

#### 3.2.4. Effect of interlayer interface treatment on shear-bearing capacity

The combined beams with the same other factors but different interface treatments between the ECC and concrete were divided into groups, and the cracking load and ultimate load of 13 groups of beams were compared, as shown in Figure 14. The vertical coordinate is the shear strength v = V/(bh0), and the horizontal coordinate is the beam group number. It can be seen that the interface treatment between the ECC and concrete has no significant effect on the cracking load and ultimate load of the beams [34].

## 4. Finite Element (FE) Analysis

### 4.1. Modeling

#### 4.1.1. Model Overview

The available FE analysis software ABAQUS was used to build the FE analysis model of the combined ECC–concrete beam. Concrete, ECC material, and mat are selected as C3D8 units. The test beams are supported beams, and the FE model is set up with the same boundary conditions as the test, with the left end support point set as a hinge support and the right end support point set as a sliding support, as shown in Figure 15.

#### 4.1.2. Material Ontology Model

In order to simplify the analysis, the principal reinforcement model is an ideal elastoplastic model with the modulus of elasticity and Poisson’s ratio of 200 GPa and 0.3, respectively. The principal model of concrete recommended in the Code for the Design of Concrete Structures, GB 50010-2010 [35], is used, and the stress-strain curves in tension and compression are shown in Figure 16. The dimensional system used in the modeling process in this paper is as shown in Table 10.

The expression of the compressive stress-strain relationship curve is as follows:(6)$$\sigma =(1-d_{c})E_{c}\varepsilon$$
(7)$$d_{c}=\left\{\begin{array}{l} 1-\frac{\rho_{c}n}{n-1+x^{n}}\\ 1-\frac{\rho_{c}}{\alpha_{c}{\left(x-1\right)}^{2}+x} \end{array}\right.$$
(8)$$x=\frac{\varepsilon }{\varepsilon_{\mathrm{cr}}}$$
(9)$$\rho_{c}=\frac{f_{\mathrm{cr}}}{E_{c}\varepsilon_{\mathrm{cr}}}$$
(10)$$n=\frac{E_{c}\varepsilon_{\mathrm{cr}}}{E_{c}\varepsilon_{\mathrm{cr}}-f_{\mathrm{cr}}}$$
where $\alpha_{t},\;\alpha_{c}$ are the parameters of the descending section of the uniaxial tension and compression stress-strain curves of concrete, and their reference values are adopted in accordance with the specifications of the Code for the Design of Concrete Structures, GB 50010-2010 [35]; $f_{\mathrm{tr}},\;f_{\mathrm{cr}}$ are the concrete uniaxial tensile parameters, respectively; $\varepsilon_{\mathrm{tr}},\;\varepsilon_{\mathrm{cr}}$ denote the peak concrete strains corresponding to $f_{\mathrm{tr}},\;f_{\mathrm{cr}}$; and $d_{t},\;d_{c}$ are the concrete uniaxial tensile and compressive damage parameters, respectively.

The ECC tensile principal structure model of Han [36] is used in this paper, as shown in Figure 17a, combined with the various data collected during the experiments in this paper, with the following expressions:(11)$$\sigma (\varepsilon )=\left\{\begin{array}{c} E_{0}\varepsilon \;\;\;\;\;\;\;\;\;\;\;\;\;\;\;\;\;\;\;\;\;\;\;\;\;\;\;\;\;\;\;\;\;\;\;\;\;0\leq \varepsilon \;<\varepsilon_{\mathrm{cr}}\\ \sigma_{\mathrm{cr}}+(\sigma_{\mathrm{tp}}-\sigma_{\mathrm{cr}})\frac{\varepsilon -\varepsilon_{\mathrm{cr}}}{\varepsilon_{\mathrm{tp}}-\varepsilon_{\mathrm{cr}}}\;\;\;\;\;\;\varepsilon_{\mathrm{cr}}\leq \varepsilon \;<\varepsilon_{\mathrm{tp}}\\ \sigma_{\mathrm{tp}}-\sigma_{\mathrm{tp}}\frac{\varepsilon -\varepsilon_{\mathrm{tp}}}{\varepsilon_{\mathrm{tu}}-\varepsilon_{\mathrm{tp}}}\;\;\;\;\;\;\;\;\;\;\;\;\;\;\;\;\;\;\;\varepsilon_{\mathrm{tp}}\leq \varepsilon \;<\varepsilon_{\mathrm{tu}} \end{array}\right.$$
where $E_{0}$ is the initial modulus of elasticity in the elastic phase of ECC; $\sigma_{\mathrm{cr}}$, $\sigma_{\mathrm{tp}}$ are the initial crack and peak stress, respectively; and $\varepsilon_{\mathrm{cr}}$, $\varepsilon_{\mathrm{tp}}$, and $\varepsilon_{\mathrm{tp}}$ are the initial crack strain, peak strain, and ultimate strain, respectively.

The ECC compressive principal structure model proposed by Maalej et al. [37] is used, as shown in Figure 17b, with the following expressions,
(12)$$Y=\left\{\begin{array}{c} a_{0}+a_{1}X+a_{2}X+a_{3}X^{3}\;\;\;\;\;\;0\leq X\leq 1\\ \frac{X}{b_{0}{\left(X-1\right)}^{2}+X}\;\;\;\;\;\;\;\;\;\;\;\;\;\;\;\;X\geq 1 \end{array}\right.$$
(13)$$X=\varepsilon /\varepsilon_{\mathrm{peak}},Y=\sigma /\sigma_{\mathrm{peak}}$$
where $\sigma_{\mathrm{peak}}$ and $\varepsilon_{\mathrm{peak}}$ are the peak stress and strain of the test results, respectively; and a_0_, a_1_, a_2_, a_3_, and b_0_ are fixed constants with fitted values of 0.003, 0.8, 0.66, −0.395, and 43.404, respectively.

### 4.2. Load-Deflection Curve Analysis

The load-span displacement curves of each beam obtained from the FE calculations were taken and compared with the experimental results, as shown in Figure 18. It can be seen that the deflection curves of the test and FEs follow the same trend. However, the deflection curve of the FE simulation is slightly stiffer than the test value, which may arise because (1) the FE simulation is idealized compared with the test, with boundary conditions and contact relations defaulted to ideal conditions, whereas the test is influenced by objective conditions; (2) the bond slip between the reinforcement, concrete, and ECC is not considered in the FE model. The curve change trend is consistent with the FE model, and the simulation achieves reasonable expectations.

### 4.3. Comparative Analysis of Load-Bearing Capacity Values

Table 11 shows the comparison of the FE analysis results and tests. The mean value of the ratio of the ultimate load test value to the calculated value is 1.09, and the coefficient of variation is 0.08, which means that the calculated value is in good agreement with the test value. The mean value of the ratio of the cracking load test value to the calculated value is 1.15, and the coefficient of variation is 0.14, which means that the deviation of individual data is large [38]. The reason for this is that the first crack observed by the naked eye was used as the basis for identifying the cracked beam in the test process; although the beam may be cracked at a certain load level [39], the crack is too small to be identified until the load increases to the extent that the crack is visible to the naked eye, resulting in the measured cracked load being large [40]. Overall, the FE simulations achieved reasonable expectations.

## 5. Conclusions

In this paper, the interface treatment, shear-to-span ratio, concrete strength, and ECC thickness are used as parameters to design 36 experimental studies on the shear resistance of ECC–concrete combination beams without web reinforcement, and the following basic conclusions are obtained through comparative studies with Abaqus FE simulation software:(1)From the damage characteristics, the damage type of the combined beam is similar to RC in the same shear-to-span ratio range, but its shear toughness, ability to maintain the integrity of the member, and shear-bearing capacity are improved. The shear-to-span ratio is an important factor affecting the shear-bearing capacity of the member, and with the increase in the shear-to-span ratio, the decrease in the shear force of the combined beam slows down after λ is greater than 2.(2)In the moderate range, ECC thickness can improve the shear performance; the interface treatment of wire mesh and groove has no significant effect on the cracking load and ultimate load size but does increase the experimental complexity and the actual construction difficulty.(3)From the load-deflection curve, the Abaqus simulation results are slightly stiffer than the test results. However, considering the errors brought by external factors in the test conditions and the idealization of the FE simulation by ignoring the bond-slip between materials and the FE software simulation, the error is within a reasonable range; thus, the FE analysis results predicting the ultimate load capacity and deflection are more reasonable.

## Figures and Tables

**Figure 1 materials-16-05706-f001:**
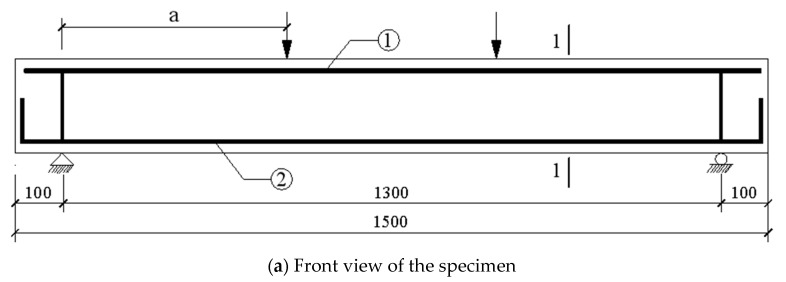
Geometry and construction of the specimen (unit: mm).

**Figure 2 materials-16-05706-f002:**
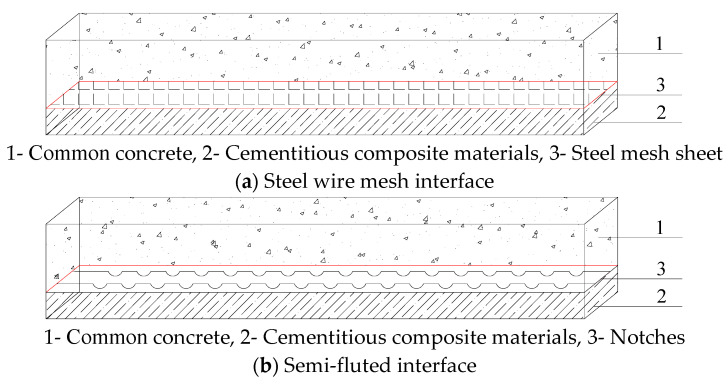
Interstorey interface treatment for modular beams.

**Figure 3 materials-16-05706-f003:**
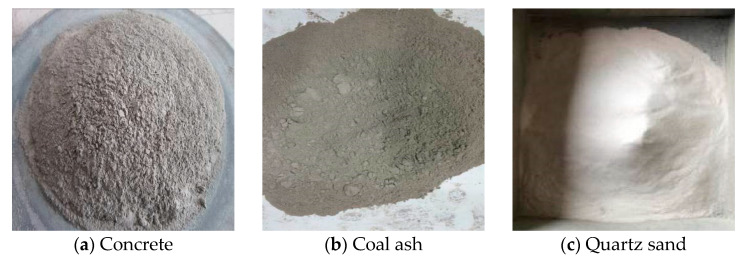
Test materials.

**Figure 4 materials-16-05706-f004:**
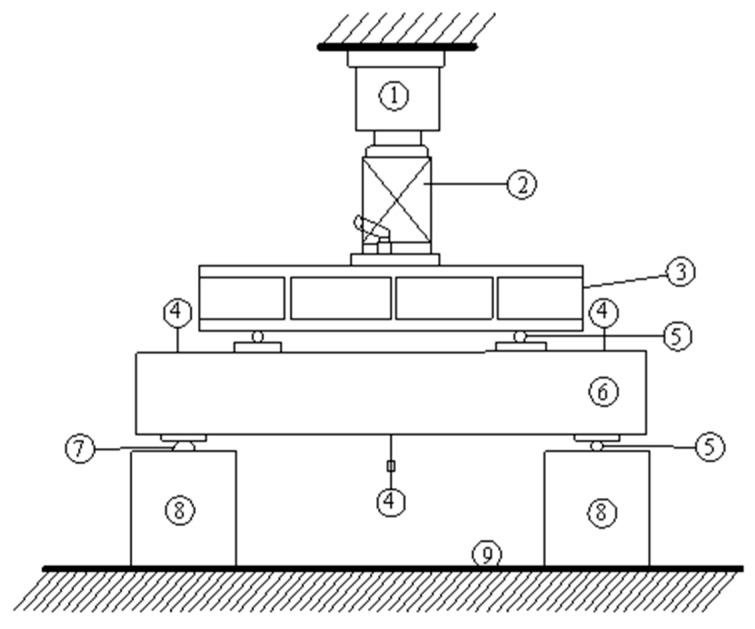
Schematic diagram of the loading device: 1. force sensor; 2. jack; 3. distribution beam; 4. percentage table; 5. sliding support; 6. test beam; 7. hinge support; 8. pier; 9. test bench.

**Figure 5 materials-16-05706-f005:**
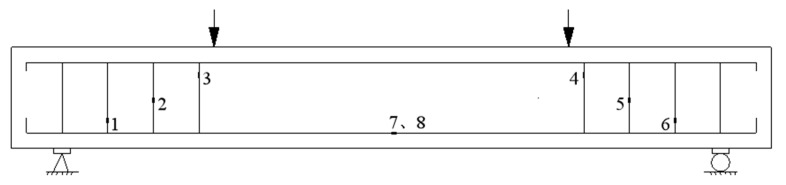
Strain gauge positions.

**Figure 6 materials-16-05706-f006:**
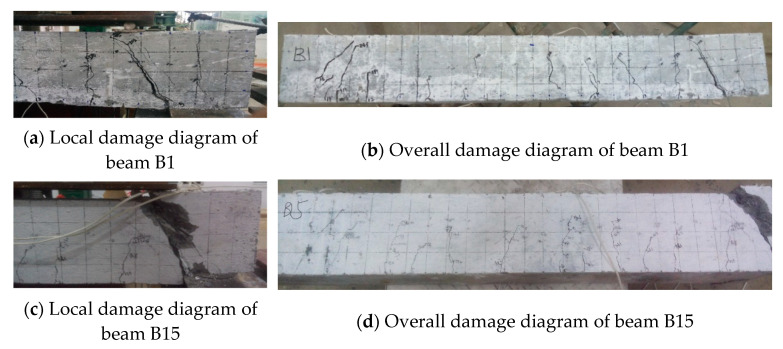
Beams B1, B15 damage diagram.

**Figure 7 materials-16-05706-f007:**
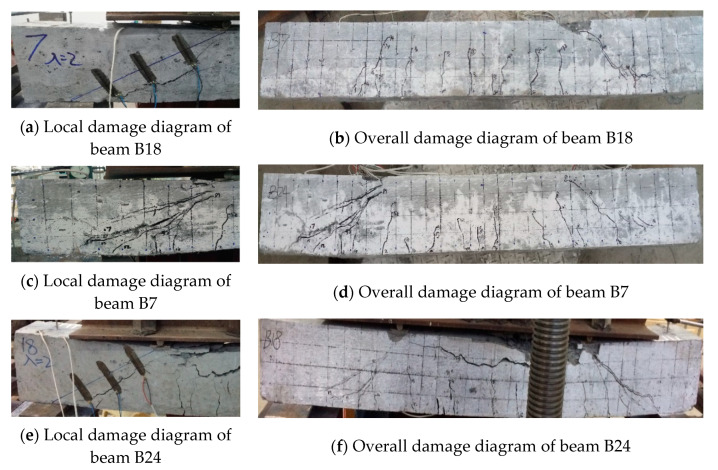
Beams B18, B7, B24 damage diagram.

**Figure 8 materials-16-05706-f008:**
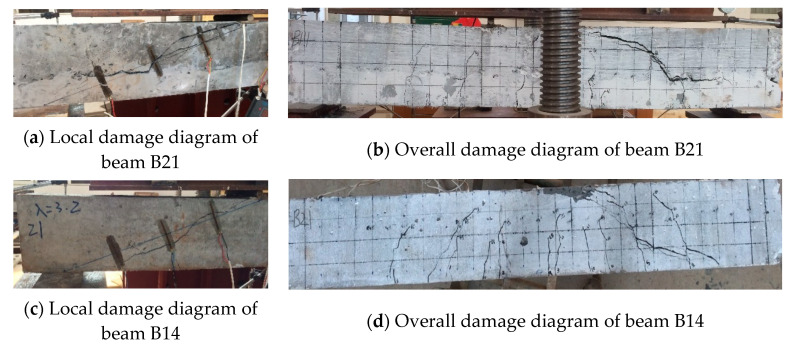
Beams B21, B14 damage diagram.

**Figure 9 materials-16-05706-f009:**
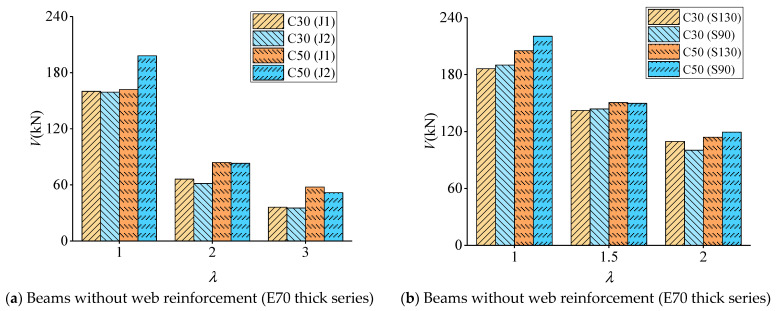
Effect of concrete strength on shear load resistance of concrete–ECC beams.

**Figure 10 materials-16-05706-f010:**
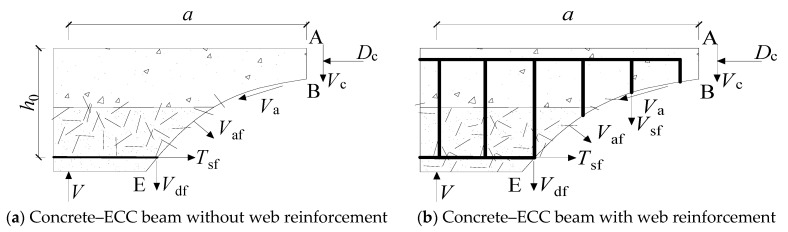
Concrete–ECC beam isolation body force mechanism.

**Figure 11 materials-16-05706-f011:**
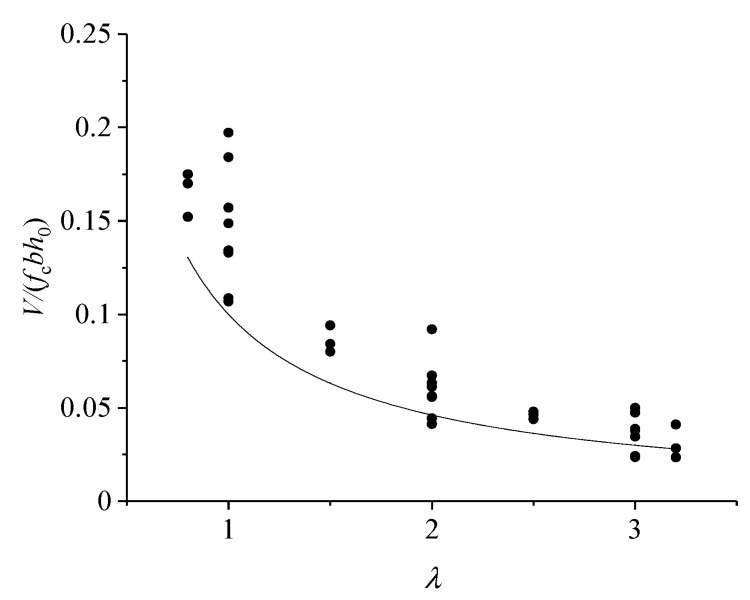
Relationship between *V*/(*f*_c_*bh*_0_*)* and λ.

**Figure 12 materials-16-05706-f012:**
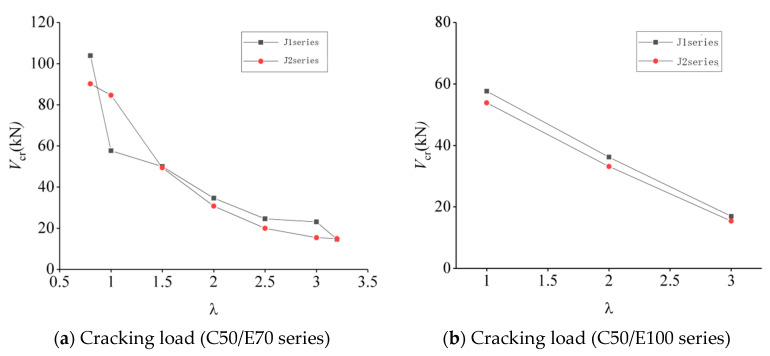
Effect of shear-to-span ratio on shear load resistance of concrete–ECC beams.

**Figure 13 materials-16-05706-f013:**
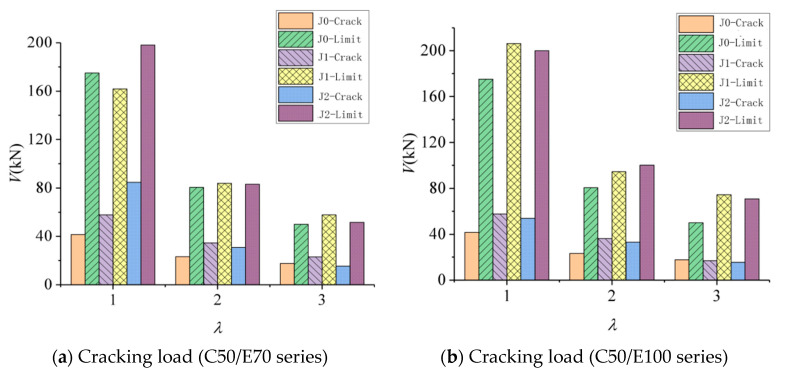
Effect of ECC thickness on shear load resistance of concrete–ECC beams.

**Figure 14 materials-16-05706-f014:**
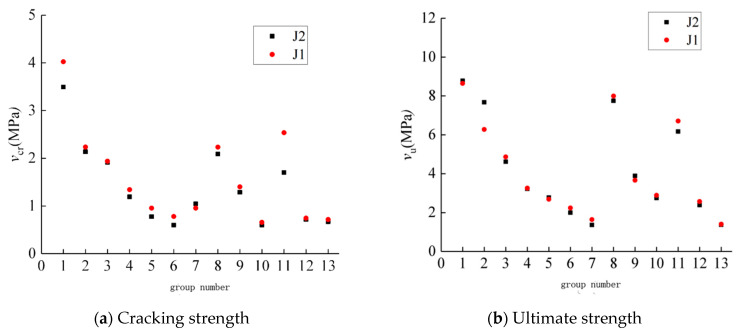
Effect of interface form on shear strength of concrete–ECC beams without webs.

**Figure 15 materials-16-05706-f015:**
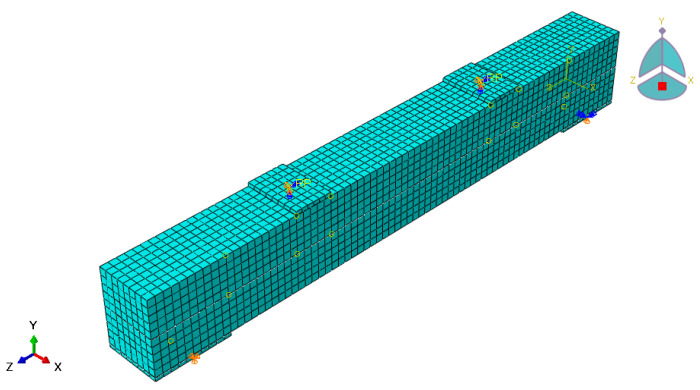
Finite element model.

**Figure 16 materials-16-05706-f016:**
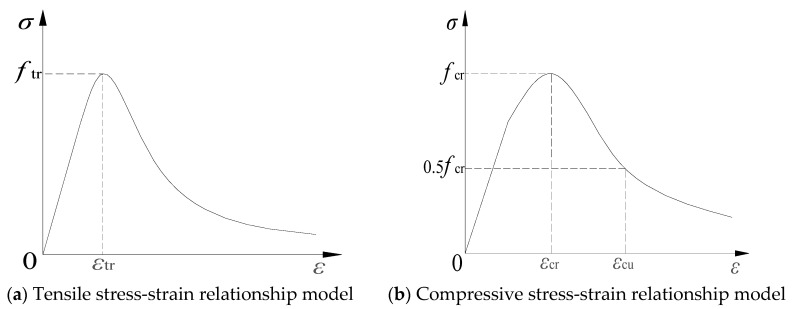
Concrete stress-strain principal structure model.

**Figure 17 materials-16-05706-f017:**
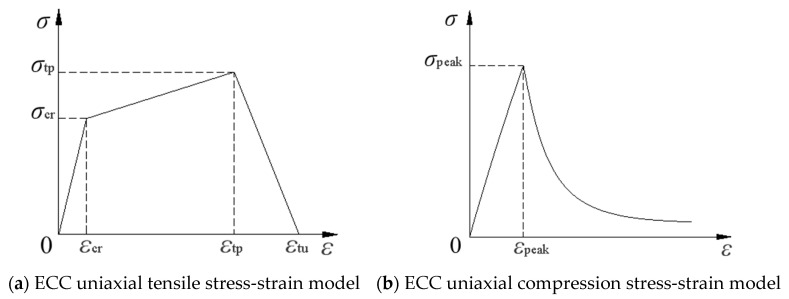
ECC stress-strain principal structure model.

**Figure 18 materials-16-05706-f018:**
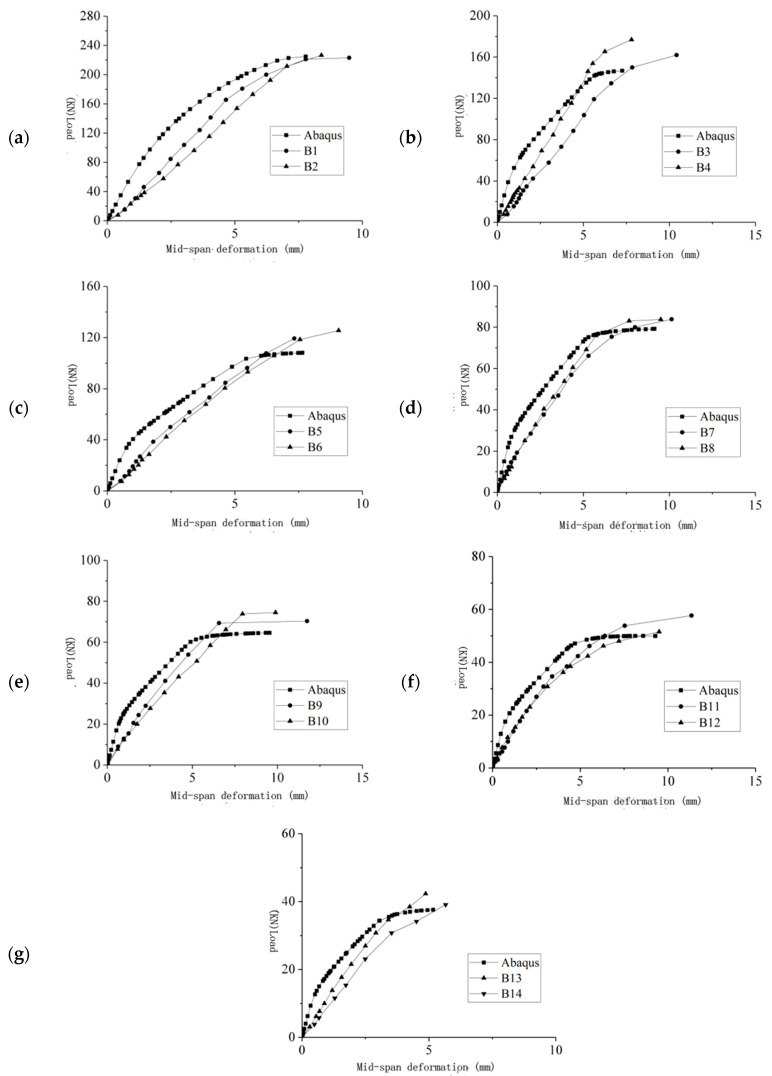
(**a**–**g**) Load-span deformation in this curve.

**Table 1 materials-16-05706-t001:** 1. 70ECC thickness beams. 2 100ECC thickness beams. 3 C30 strength beams.

Number	Interface	Strength	Shear-to-Span Ratio	ECC Thickness (mm)	Number	Interface	Strength	Shear-to-Span Ratio	ECC Thickness (mm)
1									
B1	J1	C50	0.8	70	B2	J2	C50	0.8	70
B3	J1	C50	1	70	B4	J2	C50	1	70
B5	J1	C50	1.5	70	B6	J2	C50	1.5	70
B7	J1	C50	2	70	B8	J2	C50	2	70
B9	J1	C50	2.5	70	B10	J2	C50	2.5	70
B11	J1	C50	3	70	B12	J2	C50	3	70
B13	J1	C50	3.2	70	B14	J2	C50	3.2	70
B15	J0	C50	0.8	0	B16	J0	C50	1	0
B17	J0	C50	1.5	0	B18	J0	C50	2	0
B19	J0	C50	2.5	0	B20	J0	C50	3	0
B21	J0	C50	3.2	0	B22	J1	C30	1	70
B23	J2	C30	1	70	B24	J1	C30	2	70
B25	J2	C30	2	70	B26	J1	C30	3	70
B27	J2	C30	3	70	B28	J0	C30	1	0
B29	J0	C30	2	0	B30	J0	C30	3	0
2									
B1	J1	C50	1	100	B2	J2	C50	1	100
B3	J1	C50	2	100	B4	J2	C50	2	100
B5	J1	C50	3	100	B6	J2	C50	3	100
3									
B1	J1	C50	1	100	B2	J2	C50	1	100
B3	J1	C50	2	100	B4	J2	C50	2	100
B5	J1	C50	3	100	B6	J2	C50	3	100
B23	J2	C30	1	70	B24	J1	C30	2	70
B25	J2	C30	2	70	B26	J1	C30	3	70
B27	J2	C30	3	70	B28	J0	C30	1	0
B29	J0	C30	2	0	B30	J0	C30	3	0

**Table 2 materials-16-05706-t002:** Main parameters of cement.

Items		National Standard (GB175-2007)	The Product
Standard consistency (%)		24~30	24
Stability (mn)		eligible	eligible
Fineness (80 µm sieve allowance/%)		≤10	5
MgO/SO_3_/CL^−^		≤5.0/≤3.5/≤0.06	1.21/2.5/0.01
Alkali content (%)		≤0.6	0.45
Solidification time (min)	Incipient condensation time	≥45	210
	Time of final coagulation	≤600	266
Flexural strength (MPa)	3 d	≥3.5	6.07
	28 d	≥6.5	9.38
Compressive strength (MPa)	3 d	≥17.0	30.93
	28 d	≥42.5	48.42

**Table 3 materials-16-05706-t003:** Fly ash technical parameters.

SiO_2_ (%)	Al_2_O_3_ (%)	Fe_2_O_3_ (%)	CaO (%)	MgO (%)
49.54	33.2	4.54	6.1	3.2

**Table 4 materials-16-05706-t004:** Technical parameters of silica fume.

SiO_2_ (%)	CaO (%)	K_2_O (%)	MgO (%)	Al_2_O_3_ (%)	P_2_O_5_ (%)
97.35	0.568	0.447	0.414	0.337	0.315

**Table 5 materials-16-05706-t005:** Quartz sand technical parameters.

Proportion (g/cm^3^)	Capacity (g/cm^3^)	Attrition Rate (%)	Porosity (%)	Mohs Hardness	Mud Content (%)	Unevenness Coefficient (k80)
2.66	1.75	0.35	43	7.5	≤1	≤1.8

**Table 6 materials-16-05706-t006:** Performance parameters of PVA fiber.

Length (mm)	Caliber (µm)	Modulus of Elasticity (GPa)	Elongation (%)	Tensile Strength (MPa)	Densities (g/cm^3^)
12	12~18	35	6–8	1200	1.3

**Table 7 materials-16-05706-t007:** Technical parameters of concrete sand.

Items	Mud Content (%)	Apparent Density (kg/m^3^)	Fineness Modulus (MX)
Results	1.2	2620	2.6

**Table 8 materials-16-05706-t008:** Technical parameters of concrete stone.

Items	Apparent Density (kg/m^3^)	Mud Content (%)	Crushing Value (%)	Needle and Flake Content (%)
Results	2800	0.5	8	4

**Table 9 materials-16-05706-t009:** ECC and concrete mix ratio.

Substrate Material	Cement	Fly Ash	Quartz Sand	Gravel	Silica Fume	Water-to-Glue Ratio	Water Reducer	PVA Volume Rate (%)
ECC	0.2	0.6	0.36	-	0.2	0.35	0.004	2
C30 concrete	1	-	1.37	2.44	-	0.4	0.003	-
C50 concrete	1	-	1.36	2.26	-	0.35	0.003	-

**Table 10 materials-16-05706-t010:** Dimensional system used in this paper.

Amount	Strength	Stresses	Lengths
unit	N	MPa	mm

**Table 11 materials-16-05706-t011:** Comparison of FE results and tests.

Beam Number	*V* _crexp_	*V* _crAbaqus_	*V*_crexp_/*V*_crAbaqus_	*V* _uexp_	*V* _uAbaqus_	*V*_uexp_/*V*_uAbaqus_
B1	103.85	87.5	1.19	223.08	224.61	0.99
B2	90.19	87.5	1.03	226.62	224.61	1.01
B3	57.69	55.47	1.04	181.93	147.98	1.23
B4	84.63	55.47	1.53	198.08	147.98	1.34
B5	50.00	45.15	1.11	125.46	108.22	1.16
B6	49.38	45.15	1.09	119.23	108.22	1.10
B7	34.62	26.67	1.30	83.85	79.21	1.06
B8	30.77	26.67	1.15	83.08	79.21	1.05
B9	24.62	21.89	1.12	69.23	64.69	1.07
B10	20.00	21.89	0.91	74.54	64.69	1.15
B11	23.08	18.68	1.24	57.69	49.99	1.15
B12	15.39	18.68	0.82	51.54	49.99	1.03
B13	14.62	12.99	1.53	42.31	37.64	1.23
B14	14.92	12.99	1.67	40.75	37.64	1.08
L1	54.75	42.15	1.30	217.56	233.87	1.01
L2	40.30	42.15	0.96	217.56	234.87	1.00
L13	53.23	45.12	1.18	205.29	170.63	1.20
L14	52.47	45.12	1.16	180.20	170.63	1.06
L15	39.54	35.23	1.12	150.55	122.48	1.23
L16	37.26	35.23	1.06	117.86	122.48	0.96
L17	31.94	27.1	1.18	114.05	104.74	1.09
L18	29.66	27.1	1.09	111.01	104.74	1.06
L41	31.18	33.12	0.94	123.18	117.25	1.05
L42	32.70	34.23	0.96	115.57	111.74	1.03

Note: *V*_cr_ denotes cracking load size, kN; the subscript “exp” denotes test; the subscript “Abaqus” denotes simulation.

## Data Availability

All relevant data are within the paper.

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
