# Peer review of "Experimental Study of the Shear Performance of Combined Concrete–ECC Beams without Web Reinforcement"

_materials, 2023, doi:10.3390/ma16165706_

Round 1
Reviewer 1 Report
The authors designed 36 mixed concrete-ECC (fiber-reinforced cementitious compound) beams and evaluated their shear resistance behavior. In addition, they propose a model through a finite element analysis. I believe that it is important that this type of research focused on improving the durability of the structures necessary in the construction industry be published.1) Is important to add information about the materials employed in the beams (properties of the aggregates, cement, fibers, etc.).
2) Determine in which consist both the wire mesh and emispherical concave surface at the concrete-ECC interlayer interface, and what is the form in which are incorporated at the beams
3) In the figure 1 a), b) and c), what means the points in the beams, that are describing by a circle with the numbers 1 and 2?
4) To describe in better form the experimental procedure, the table 1, could be separate in 3 differents tables; in the 1st table would only incluye the beams with 70 ECC thickness (J1 and J2) and the control beams (JO); in the second table would be only incluye the beams with 100 ECC thickness; and 3rd table would be incluyed the beams with C30 strength, additionally incluid in the table 3 the 100 ECC thickness beams
5) Why when the shear-to-span ratio is moderate, in the analysis mix 70 and 100 ECC thickness and, not only the analysis is make with 70 ECC thickness?
6) Why when the shear-to-span ratio is relatively large, is make only analysis both B21 and B14 beams?, is recommended to include the B13 beam in the analysis. 7) Indicate in the figure 7, that analysis is only refered to 70 ECC thickness. What will be the responce to shear load resistance with 100 ECC thickness beams?
8) Is not clear the increase of 32% mentioned in the line 201.
9) Is not clear what is the impact in the results the C30 beams, because the results of this beams were not incluyed in the apart 3.2. Concrete-ECC shear bearing capacity analysis
10) In the figure 14, the nomenclature is in another idioma
11) In both the experimental overview and test results and analysis, the autors not incluyed some references which support your manuscripst
12) In general, is necessary to improve the redaction, because the use of the punctuations is not correct.
The writing should be improved, in some part of the manuscript it´s not easy to understand.
Author Response
Response to Reviewers' Comments (Manuscript ID materials-2471297)
Experimental study of the shear performance of combined concrete-ECC beams without web reinforcement
Author's Response
Dear Editors and Reviewers,
Thank you for allowing us to submit a revised version of the manuscript "Experimental study of the shear performance of combined concrete-ECC beams without web reinforcement" for publication in the Scientific Reports. We sincerely appreciate the time and effort you and the reviewers dedicated to providing feedback on our manuscript and are grateful for our paper's insightful and constructive comments. The thorough review helped immensely in shaping and improvement of the manuscript. We have incorporated most of the suggestions made by the reviewers. Please see below; Fonts in blue have been provided for a point-by-point response to the reviewers' comments and concerns. Also, it should be noted that all of the editors' and reviewers' suggestions have been applied to the paper in addition to the reviewer's comments.
Reviewers' comments:
Reviewer #1:
The authors designed 36 mixed concrete-ECC (fiber-reinforced cementitious compound) beams and evaluated their shear resistance behavior. In addition, they propose a model through a finite element analysis. I believe that it is important that this type of research focused on improving the durability of the structures necessary in the construction industry be published.
Authors' Response: Thank you for your kind remarks. We will make every effort to revise the document.
- Is important to add information about the materials employed in the beams (properties of the aggregates, cement, fibers, etc.).
Authors’ Response: Thank you for your constructive comment. The authors have added the properties of important materials including aggregates, cement, fibers, etc. in section 2.2. and highlighted in red in the paper. Thanks.
- Determine in which consist both the wire mesh and emispherical concave surface at the concrete-ECC interlayer interface, and what is the form in which are incorporated at the beams
Authors’ Response: We thank the reviewer for these questions. To make it easier to answer this question, the authors more down the chart in the paper.
(a) steel wire mesh interface |
(b) semi-fluted interface Figure2. Interstorey interface treatment for modular beams |
- In the figure 1 a), b) and c), what means the points in the beams, that are describing by a circle with the numbers 1 and 2?
Authors’ Response: Thank you for pointing these out. In the figure 1 a), b) and c), The dots on the beams described by circles with the numbers 1 and 2 mean auxiliary reinforcement and tension reinforcement respectively.
- To describe in better form the experimental procedure, the table 1, could be separate in 3 differents tables; in the 1st table would only incluye the beams with 70 ECC thickness (J1 and J2) and the control beams (JO); in the second table would be only incluye the beams with 100 ECC thickness; and 3rd table would be incluyed the beams with C30 strength, additionally incluid in the table 3 the 100 ECC thickness beams
Authors' Response: Thank you for pointing these out. We've divided the original form into three new forms as you said.
- Why when the shear-to-span ratio is moderate, in the analysis mix 70 and 100 ECC thickness and, not only the analysis is make with 70 ECC thickness?
Authors’ Response: Thank you for pointing these out. This study presents the design of two different thicknesses of ECC, namely 70mm and 100mm. These thicknesses correspond to the replacement height of the ECC layer for both cases.
Figure 12 illustrates the impact of the thickness of the ECC layer on the shear capacity of the combined beams.Concrete-ECC beams have varying degrees of improved shear capacity as compared to reinforced concrete beams. The shear capacity of beams without web rises with the increase in shear span ratio at an ECC layer thickness of 70 mm, in comparison to RC beams. The shear capacity is significantly increased when the thickness of the ECC layer is 100mm. The substitution of conventional concrete with ECC in the tensile zone of a beam leads to enhanced tensile and shear characteristics in the lower region. This compensates for the inherent weaknesses of brittle concrete, which is susceptible to cracking and failure in the lower portion. Consequently, this substitution is advantageous for improving overall stress performance. However, it should be noted that steel reinforcement also possesses excellent tensile properties. Therefore, it is not necessary to increase the thickness of the ECC beyond what is optimal.
Figure 12 illustrates the impact of the thickness of the ECC layer on the shear capacity of the combined beams. The comparison of shear capacity was conducted by using ECC thicknesses of 70 and 100, as shown in Figure 12.
|
|
(a) Cracking load (C50/E70 series) |
(b) Cracking load (C50/E100 series) |
Figure 12. Effect of ECC thickness on shear load resistance of concrete-ECC beams |
The following is added to the paper: The results of the present study show that, in comparison to RC beams subjected to the same conditions, the average increase in shear capacity for beams with a combination of web reinforcement within the design shear span ratio is 6%. Additionally, the corre-sponding increase in cracking load is 26%. It is worth noting that beams with a hoop ratio of 0.29% exhibit a larger increase in the thickness of the ECC layer. Specifically, when the ECC layer thickness is 100 mm, the mean increase is 9%, whereas for a thickness of 70 mm, the mean increase is 6%. Conversely, beams with a hoop ratio of 0.42% only experi-ence a 2% increase when the ECC layer thickness is 70 mm, resulting in a 6% increase. The experimental results indicate that the increase in the measured value is 6% for an ECC thickness of 70mm, while the mean increase in the measured value is only 2% for an ECC thickness of 100mm. The findings indicate that the augmentation of shear in beams with web reinforcement is more pronounced when the thickness of ECC is increased, particu-larly at lower hoop rates. However, as the hoop rate increases, the impact of increasing ECC thickness on enhancing shear capacity diminishes, and in some cases, the shear ca-pacity of certain beams slightly decreases. In terms of cracking load, the outcomes for beams with and without web reinforcement exhibit similar trends.
- Why when the shear-to-span ratio is relatively large, is make only analysis both B21 and B14 beams?, is recommended to include the B13 beam in the analysis. 7) Indicate in the figure 7, that analysis is only refered to 70 ECC thickness. What will be the responce to shear load resistance with 100 ECC thickness beams?
Authors’ Response: Thank you for pointing these out. (1) In response to the question, " Why when the shear-to-span ratio is relatively large, is make only analysis both B21 and B14 beams?, is recommended to include the B13 beam in the analysis.?" . In fact, the test analyzed all the specimens, and for lack of space, only B21 and B14 beams were listed. The test diagrams are shown below:
|
|
(a) Local damage diagram of beam B1 |
(b) B1 beam overall damage diagram |
|
|
(c) Local damage diagram of beam B15 |
(d) Overall damage diagram of beam B15 |
Figure 6. Beam B1, B15 damage diagram |
|
|
(a) Local damage diagram of beam B18 |
(b) Overall damage diagram of beam B18 |
|
|
(c) Local damage diagram of beam B7 |
(d) Overall damage diagram of beam B7 |
|
|
(e) Local damage diagram of beam B24 |
(f) Overall damage diagram of beam B24 |
Figure 7. Beam B18, B7, B24 damage diagram |
|
|
(a) Local damage diagram of beam B21 |
(b) Overall damage diagram of beam B21 |
|
|
(c) Local damage diagram of beam B14 |
(d) Overall damage diagram of beam B14 |
Figure 8. Beam B21, B14 damage diagram |
(2) For the question“7) Indicate in the figure 7, that analysis is only refered to 70 ECC thickness. What will be the responce to shear load resistance with 100 ECC thickness beams?”
In fact, the test analyzed all the specimens, and for the sake of space, the analysis only refers to the 70 ECC thickness. The authors have modified Figure 7 by adding the "Beam with web reinforcement (J1E70 thick series)" case diagram.
In this paper, two ECC thicknesses of 70mm and 100mm are designed, and the enhancement effect is no longer obvious when the ECC thickness exceeds 70mm. Existing literature [27] on the concrete-ECC combination beam flexural test shows that when the ECC thickness accounts for more than 0.2H of the section height, the ECC enhancement effect is no longer obvious. Further analysis shows that the combined effect of ECC thickness and hoop rate is better than that of 100 thickness when 70 mm thickness is used, and the best effect is achieved when the hoop rate is 0.29%.
Changes in Manuscript:
3.2.1. The effect of concrete strength on shear bearing capacity
The effect of concrete strength in the compression zone on the shear strength of concrete-ECC beams is shown in Figure 9, where the vertical coordinate is the shear bearing capacity V, and the horizontal coordinate is the shear-to-span ratio λ.
It should be noted that Figure 9 illustrates the impact of concrete strength on shear loads in concrete-ECC beams. This serves as an illustrative example for beams without web reinforcement (E70 thick series) and beams with web reinforcement (J1E70 thick series).
Figure 7 illustrates the impact of concrete strength on the combined beams lacking web reinforcement. Notably, this effect is more prominent in the concrete-engineered ECC beams compared to the RC beams. Specifically, the average increase in shear strength for the beams lacking web reinforcement is 32% when the concrete strength grade is elevated from C30 to C50. In contrast, the average increase in shear strength for the beams with web reinforcement is 10%. Furthermore, it has been shown that the enhancement in the shear strength of beams has a diminishing trend as the shear span ratio increases, particularly when the concrete strength class transitions from C30 to C50. One possible explanation for this phenomenon is that as the shear span ratio increases, the damage type of the beam transitions from inclined compression damage to inclined tensile damage. This transition occurs due to changes in the internal force transmission within the beam. Consequently, the influence of material strength on the shear capacity of the beam also changes[22, 23].
|
|
Figure 9. Effect of concrete strength on shear load resistance of concrete-ECC beams |
For instance, the compressive strength of the concrete significantly affects the shear strength of the beam during inclined compression damage. Conversely, the tensile properties of the material in the tension zone play a crucial role in the occurrence of inclined tensile damage in the beam. In the context of diagonal compression damage in beams, the influence of concrete compressive strength on beam shear strength is observed to be more pronounced. Conversely, in the case of diagonal tension damage in beams, the tensile properties of the material within the tensile zone exhibit an augmented role, leading to a corresponding decrease in the impact of concrete compressive strength[24, 25].
When comparing ECC material to concrete, it is seen that ECC material does not exhibit significant advantages in terms of compressive performance. However, it does possess better tensile qualities, making it appropriate for implementation in the tensile zone of beams. This application serves to enhance the shear brittleness of the beam[26].
To examine the shear behavior of concrete-ECC beams, Figure 9 illustrates the force diagram of concrete-ECC beams subjected to shear. The diagram includes various forces: Vcf, representing the shear force provided by ECC; Dc, which denotes the compressive stress in the shear-compression zone and is analogous to the compressive properties of ECC and concrete observed in RC beams; Vaf, indicating the bridging force supplied by fibers at the diagonal cracks; Tsf, representing the combined force of the tensile reinforcement pulling force after fiber reinforcement; and Vdf, signifying the pinning force exerted by the tensile reinforcement and ECC.
|
|
Figure. 9 Concrete-ECC beam isolation body force mechanism |
Taking the concrete-ECC beam without web as an example, its shear capacity and shear span ratio can be obtained as the dimensionless parameter V/(fcbh0), and the equation of the relationship between V/(fcbh0)and the shear span ratio is
|
(1) |
The test data of the webless beams in this work were evaluated using the program Origin to fit the lower envelope of the data. The resulting relationship of k1(λ) is shown in Equation (2). The contrast between the equation represented by Eq. (2) and the experimental data is seen in Figure10. Next, the concrete shear contribution Vc was deducted from the shear capacity V to get the shear contribution Vf of ECC, which was done using the test data obtained from webless reinforced beams. The resulting connection between Vf and the shear-to-span ratio is represented by Eq. (3). The formula for calculating the shear capacity of a webless reinforced concrete-ECC combination beam may be expressed as Eq. (4), more precisely Eq. (5).
|
(2) |
|
|
(3) |
|
V=Vc+Vf |
(4) |
|
|
(5) |
|
In this study, the impact coefficient of the fiber reinforcing material is denoted as ' '. Two different thicknesses of ECC, namely 70mm and 100mm, were designed and evaluated. It was shown that the augmentation effect diminishes beyond an ECC thickness of 70mm. The available scholarly literature [27] pertaining to the beam bending test of concrete-ECC combination indicates that the increase provided by ECC becomes less significant when the thickness of ECC exceeds 0.2H. Based on the findings presented in this study, it is seen that the calculation outcomes exhibit greater levels of satisfaction when =0.35. Upon further examination, it is evident that the thorough evaluation of ECC thickness and hoop rate demonstrates superior outcomes when using a thickness of 70mm, as opposed to 100mm. Moreover, the most optimal outcome is achieved when the hoop rate is 0.29%.
|
Figure 10. Relationship between V/(fcbh0) and λ |
- Is not clear the increase of 32% mentioned in the line 201.
Authors’ Response: 32% means "The average increase in shear strength of beams without web reinforcement is 32% when the concrete strength class is increased from C30 to C50,".
- Is not clear what is the impact in the results the C30 beams, because the results of this beams were not incluyed in the apart 3.2. Concrete-ECC shear bearing capacity analysis
Authors’ Response: Thanks for reminding that. The author has modified the content of section "3.2.1.". Added a discussion of "Results for C30 beams."
Compared to RC beams, for concrete-ECC beams, the effect of concrete strength on the combined beams without web reinforcement is more pronounced, as the mean value of the increase in shear strength of the beams without web reinforcement is 32% when the concrete strength class is increased from C30 to C50, while the mean value of the increase in shear strength of the beams with web reinforcement is 10%. In addition, when the concrete strength class is increased from C30 to C50, the increase in shear strength of beams tends to decrease with the increase in shear span ratio.
- In the figure 14, the nomenclature is in another idioma
Authors’ Response: The reviewer made a great point. The gauge system used in the modeling process in this paper is as follows:
Table 10 Dimensional systemused in this paper
Amount |
Strength |
|
Stresses |
Lengths |
unit |
N |
|
MPa |
mm |
- In both the experimental overview and test results and analysis, the autors not incluyed some references which support your manuscripst
Authors’ Response: Thank you for your constructive comment. writers have included relevant references to substantiate the testing and analysis conducted in this research article.
- In general, is necessary to improve the redaction, because the use of the punctuations is not correct.
Authors’ Response: Thank you for pointing this out. The authors diligently evaluate and update the whole of the content, including aspects such as language, formatting, and punctuation, among others, in order to meet the expectations of the reviewers. This article aims to provide a comprehensive and intelligible analysis of the subject matter, with the intention of conveying valuable information to the reader.
Response to Reviewers' Comments (Manuscript ID materials-2471297)
Experimental study of the shear performance of combined concrete-ECC beams without web reinforcement
Author's Response
Dear Editors and Reviewers,
Thank you for allowing us to submit a revised version of the manuscript "Experimental study of the shear performance of combined concrete-ECC beams without web reinforcement" for publication in the Scientific Reports. We sincerely appreciate the time and effort you and the reviewers dedicated to providing feedback on our manuscript and are grateful for our paper's insightful and constructive comments. The thorough review helped immensely in shaping and improvement of the manuscript. We have incorporated most of the suggestions made by the reviewers. Please see below; Fonts in blue have been provided for a point-by-point response to the reviewers' comments and concerns. Also, it should be noted that all of the editors' and reviewers' suggestions have been applied to the paper in addition to the reviewer's comments.
Reviewers' comments:
Reviewer #1:
The authors designed 36 mixed concrete-ECC (fiber-reinforced cementitious compound) beams and evaluated their shear resistance behavior. In addition, they propose a model through a finite element analysis. I believe that it is important that this type of research focused on improving the durability of the structures necessary in the construction industry be published.
Author's Response: Thank you for your kind remarks. We will make every effort to revise the document.
- Is important to add information about the materials employed in the beams (properties of the aggregates, cement, fibers, etc.).
Author’s Response: Thank you for your constructive comment. The authors have added the properties of important materials including aggregates, cement, fibers, etc. in section 2.2. and highlighted in red in the paper. Thanks.
- Determine in which consist both the wire mesh and emispherical concave surface at the concrete-ECC interlayer interface, and what is the form in which are incorporated at the beams
Author’s Response: We thank the reviewer for these questions. To make it easier to answer this question, the authors more down the chart in the paper.
(a) steel wire mesh interface |
(b) semi-fluted interface Figure2. Interstorey interface treatment for modular beams |
- In the figure 1 a), b) and c), what means the points in the beams, that are describing by a circle with the numbers 1 and 2?
Author’s Response: Thank you for pointing these out. In the figure 1 a), b) and c), The dots on the beams described by circles with the numbers 1 and 2 mean auxiliary reinforcement and tension reinforcement respectively.
- To describe in better form the experimental procedure, the table 1, could be separate in 3 differents tables; in the 1st table would only incluye the beams with 70 ECC thickness (J1 and J2) and the control beams (JO); in the second table would be only incluye the beams with 100 ECC thickness; and 3rd table would be incluyed the beams with C30 strength, additionally incluid in the table 3 the 100 ECC thickness beams
Author's Response: Thank you for pointing these out. We've divided the original form into three new forms as you said.
- Why when the shear-to-span ratio is moderate, in the analysis mix 70 and 100 ECC thickness and, not only the analysis is make with 70 ECC thickness?
Author’s Response: Thank you for pointing these out. This study presents the design of two different thicknesses of ECC, namely 70mm and 100mm. These thicknesses correspond to the replacement height of the ECC layer for both cases.
Figure 12 illustrates the impact of the thickness of the ECC layer on the shear capacity of the combined beams.Concrete-ECC beams have varying degrees of improved shear capacity as compared to reinforced concrete beams. The shear capacity of beams without web rises with the increase in shear span ratio at an ECC layer thickness of 70 mm, in comparison to RC beams. The shear capacity is significantly increased when the thickness of the ECC layer is 100mm. The substitution of conventional concrete with ECC in the tensile zone of a beam leads to enhanced tensile and shear characteristics in the lower region. This compensates for the inherent weaknesses of brittle concrete, which is susceptible to cracking and failure in the lower portion. Consequently, this substitution is advantageous for improving overall stress performance. However, it should be noted that steel reinforcement also possesses excellent tensile properties. Therefore, it is not necessary to increase the thickness of the ECC beyond what is optimal.
Figure 12 illustrates the impact of the thickness of the ECC layer on the shear capacity of the combined beams. The comparison of shear capacity was conducted by using ECC thicknesses of 70 and 100, as shown in Figure 12.
|
|
(a) Cracking load (C50/E70 series) |
(b) Cracking load (C50/E100 series) |
Figure 12. Effect of ECC thickness on shear load resistance of concrete-ECC beams |
The following is added to the paper: The results of the present study show that, in comparison to RC beams subjected to the same conditions, the average increase in shear capacity for beams with a combination of web reinforcement within the design shear span ratio is 6%. Additionally, the corre-sponding increase in cracking load is 26%. It is worth noting that beams with a hoop ratio of 0.29% exhibit a larger increase in the thickness of the ECC layer. Specifically, when the ECC layer thickness is 100 mm, the mean increase is 9%, whereas for a thickness of 70 mm, the mean increase is 6%. Conversely, beams with a hoop ratio of 0.42% only experi-ence a 2% increase when the ECC layer thickness is 70 mm, resulting in a 6% increase. The experimental results indicate that the increase in the measured value is 6% for an ECC thickness of 70mm, while the mean increase in the measured value is only 2% for an ECC thickness of 100mm. The findings indicate that the augmentation of shear in beams with web reinforcement is more pronounced when the thickness of ECC is increased, particu-larly at lower hoop rates. However, as the hoop rate increases, the impact of increasing ECC thickness on enhancing shear capacity diminishes, and in some cases, the shear ca-pacity of certain beams slightly decreases. In terms of cracking load, the outcomes for beams with and without web reinforcement exhibit similar trends.
- Why when the shear-to-span ratio is relatively large, is make only analysis both B21 and B14 beams?, is recommended to include the B13 beam in the analysis. 7) Indicate in the figure 7, that analysis is only refered to 70 ECC thickness. What will be the responce to shear load resistance with 100 ECC thickness beams?
Author’s Response: Thank you for pointing these out. (1) In response to the question, " Why when the shear-to-span ratio is relatively large, is make only analysis both B21 and B14 beams?, is recommended to include the B13 beam in the analysis.?" . In fact, the test analyzed all the specimens, and for lack of space, only B21 and B14 beams were listed. The test diagrams are shown below:
|
|
(a) Local damage diagram of beam B1 |
(b) B1 beam overall damage diagram |
|
|
(c) Local damage diagram of beam B15 |
(d) Overall damage diagram of beam B15 |
Figure 6. Beam B1, B15 damage diagram |
|
|
(a) Local damage diagram of beam B18 |
(b) Overall damage diagram of beam B18 |
|
|
(c) Local damage diagram of beam B7 |
(d) Overall damage diagram of beam B7 |
|
|
(e) Local damage diagram of beam B24 |
(f) Overall damage diagram of beam B24 |
Figure 7. Beam B18, B7, B24 damage diagram |
|
|
(a) Local damage diagram of beam B21 |
(b) Overall damage diagram of beam B21 |
|
|
(c) Local damage diagram of beam B14 |
(d) Overall damage diagram of beam B14 |
Figure 8. Beam B21, B14 damage diagram |
(2) For the question“7) Indicate in the figure 7, that analysis is only refered to 70 ECC thickness. What will be the responce to shear load resistance with 100 ECC thickness beams?”
In fact, the test analyzed all the specimens, and for the sake of space, the analysis only refers to the 70 ECC thickness. The authors have modified Figure 7 by adding the "Beam with web reinforcement (J1E70 thick series)" case diagram.
In this paper, two ECC thicknesses of 70mm and 100mm are designed, and the enhancement effect is no longer obvious when the ECC thickness exceeds 70mm. Existing literature [27] on the concrete-ECC combination beam flexural test shows that when the ECC thickness accounts for more than 0.2H of the section height, the ECC enhancement effect is no longer obvious. Further analysis shows that the combined effect of ECC thickness and hoop rate is better than that of 100 thickness when 70 mm thickness is used, and the best effect is achieved when the hoop rate is 0.29%.
Changes in Manuscript:
3.2.1. The effect of concrete strength on shear bearing capacity
The effect of concrete strength in the compression zone on the shear strength of concrete-ECC beams is shown in Figure 9, where the vertical coordinate is the shear bearing capacity V, and the horizontal coordinate is the shear-to-span ratio λ.
It should be noted that Figure 9 illustrates the impact of concrete strength on shear loads in concrete-ECC beams. This serves as an illustrative example for beams without web reinforcement (E70 thick series) and beams with web reinforcement (J1E70 thick series).
Figure 7 illustrates the impact of concrete strength on the combined beams lacking web reinforcement. Notably, this effect is more prominent in the concrete-engineered ECC beams compared to the RC beams. Specifically, the average increase in shear strength for the beams lacking web reinforcement is 32% when the concrete strength grade is elevated from C30 to C50. In contrast, the average increase in shear strength for the beams with web reinforcement is 10%. Furthermore, it has been shown that the enhancement in the shear strength of beams has a diminishing trend as the shear span ratio increases, particularly when the concrete strength class transitions from C30 to C50. One possible explanation for this phenomenon is that as the shear span ratio increases, the damage type of the beam transitions from inclined compression damage to inclined tensile damage. This transition occurs due to changes in the internal force transmission within the beam. Consequently, the influence of material strength on the shear capacity of the beam also changes[22, 23].
|
|
Figure 9. Effect of concrete strength on shear load resistance of concrete-ECC beams |
For instance, the compressive strength of the concrete significantly affects the shear strength of the beam during inclined compression damage. Conversely, the tensile properties of the material in the tension zone play a crucial role in the occurrence of inclined tensile damage in the beam. In the context of diagonal compression damage in beams, the influence of concrete compressive strength on beam shear strength is observed to be more pronounced. Conversely, in the case of diagonal tension damage in beams, the tensile properties of the material within the tensile zone exhibit an augmented role, leading to a corresponding decrease in the impact of concrete compressive strength[24, 25].
When comparing ECC material to concrete, it is seen that ECC material does not exhibit significant advantages in terms of compressive performance. However, it does possess better tensile qualities, making it appropriate for implementation in the tensile zone of beams. This application serves to enhance the shear brittleness of the beam[26].
To examine the shear behavior of concrete-ECC beams, Figure 9 illustrates the force diagram of concrete-ECC beams subjected to shear. The diagram includes various forces: Vcf, representing the shear force provided by ECC; Dc, which denotes the compressive stress in the shear-compression zone and is analogous to the compressive properties of ECC and concrete observed in RC beams; Vaf, indicating the bridging force supplied by fibers at the diagonal cracks; Tsf, representing the combined force of the tensile reinforcement pulling force after fiber reinforcement; and Vdf, signifying the pinning force exerted by the tensile reinforcement and ECC.
|
|
Figure. 9 Concrete-ECC beam isolation body force mechanism |
Taking the concrete-ECC beam without web as an example, its shear capacity and shear span ratio can be obtained as the dimensionless parameter V/(fcbh0), and the equation of the relationship between V/(fcbh0)and the shear span ratio is
|
(1) |
The test data of the webless beams in this work were evaluated using the program Origin to fit the lower envelope of the data. The resulting relationship of k1(λ) is shown in Equation (2). The contrast between the equation represented by Eq. (2) and the experimental data is seen in Figure10. Next, the concrete shear contribution Vc was deducted from the shear capacity V to get the shear contribution Vf of ECC, which was done using the test data obtained from webless reinforced beams. The resulting connection between Vf and the shear-to-span ratio is represented by Eq. (3). The formula for calculating the shear capacity of a webless reinforced concrete-ECC combination beam may be expressed as Eq. (4), more precisely Eq. (5).
|
(2) |
|
|
(3) |
|
V=Vc+Vf |
(4) |
|
|
(5) |
|
In this study, the impact coefficient of the fiber reinforcing material is denoted as ' '. Two different thicknesses of ECC, namely 70mm and 100mm, were designed and evaluated. It was shown that the augmentation effect diminishes beyond an ECC thickness of 70mm. The available scholarly literature [27] pertaining to the beam bending test of concrete-ECC combination indicates that the increase provided by ECC becomes less significant when the thickness of ECC exceeds 0.2H. Based on the findings presented in this study, it is seen that the calculation outcomes exhibit greater levels of satisfaction when =0.35. Upon further examination, it is evident that the thorough evaluation of ECC thickness and hoop rate demonstrates superior outcomes when using a thickness of 70mm, as opposed to 100mm. Moreover, the most optimal outcome is achieved when the hoop rate is 0.29%.
|
Figure 10. Relationship between V/(fcbh0) and λ |
- Is not clear the increase of 32% mentioned in the line 201.
Author’s Response: 32% means "The average increase in shear strength of beams without web reinforcement is 32% when the concrete strength class is increased from C30 to C50,".
- Is not clear what is the impact in the results the C30 beams, because the results of this beams were not incluyed in the apart 3.2. Concrete-ECC shear bearing capacity analysis
Author’s Response: Thanks for reminding that. The author has modified the content of section "3.2.1.". Added a discussion of "Results for C30 beams."
Compared to RC beams, for concrete-ECC beams, the effect of concrete strength on the combined beams without web reinforcement is more pronounced, as the mean value of the increase in shear strength of the beams without web reinforcement is 32% when the concrete strength class is increased from C30 to C50, while the mean value of the increase in shear strength of the beams with web reinforcement is 10%. In addition, when the concrete strength class is increased from C30 to C50, the increase in shear strength of beams tends to decrease with the increase in shear span ratio.
- In the figure 14, the nomenclature is in another idioma
Author’s Response: The reviewer made a great point. The gauge system used in the modeling process in this paper is as follows:
Table 10 Dimensional systemused in this paper
Amount |
Strength |
|
Stresses |
Lengths |
unit |
N |
|
MPa |
mm |
- In both the experimental overview and test results and analysis, the autors not incluyed some references which support your manuscripst
Author’s Response: Thank you for your constructive comment. writers have included relevant references to substantiate the testing and analysis conducted in this research article.
- In general, is necessary to improve the redaction, because the use of the punctuations is not correct.
Authors’ Response: Thank you for pointing this out. The authors diligently evaluate and update the whole of the content, including aspects such as language, formatting, and punctuation, among others, in order to meet the expectations of the reviewers. This article aims to provide a comprehensive and intelligible analysis of the subject matter, to convey valuable information to the reader.
Comments on the Quality of English Language
The writing should be improved, in some part of the manuscript it´s not easy to understand.
Author’s Response: We appreciate this recommendation. The authors diligently evaluate and update the whole of the content, including aspects such as language, formatting, and punctuation, among others. This article aims to provide a comprehensive and intelligible analysis of the subject matter, to convey valuable information to the reader.

Reviewer 2 Report
The technical content of the paper is good enough but it should be improved a bit before publication. Some minor revisions are required
1. Please remove acronyms from the abstract (ECC at least) and put it in the main body of the paper. In the abstract be as much general as possible (more an explanation of the research compared to a detailed technical description). In its present form, Abstract is very short. I suggest to put some more information regarding methodology and main results.
2. Authors say “Shear resistance design has been a standard topic in structural design, and although the ductility characteristics of concrete can be increased by setting hoop reinforcement, the effect of transmitting shear force by the hoop and longitudinal reinforcement alone is less than ideal [5].” This is true. I suggest to consider also this very recent good reference DOI 10.1016/j.engstruct.2021.113483 that explains this topic in a deep way.
3. Authors say “In foreign countries” compared to what country ? The origin of the authors ? Please be more clear or change the sentence.
4. Please no need for colors in Table of a scientific paper. Please use eventually grey scale.
5. I suggest to improve quality of tables. Too big with a lot of text in different rows. Please check and modify accordingly.
Author Response
Response to Reviewers' Comments (Manuscript ID materials-2471297)
Experimental study of the shear performance of combined concrete-ECC beams without web reinforcement
Author's Response
Dear Editors and Reviewers,
Thank you for allowing us to submit a revised version of the manuscript "Experimental study of the shear performance of combined concrete-ECC beams without web reinforcement" for publication in the Scientific Reports. We sincerely appreciate the time and effort you and the reviewers dedicated to providing feedback on our manuscript and are grateful for our paper's insightful and constructive comments. The thorough review helped immensely in shaping and improvement of the manuscript. We have incorporated most of the suggestions made by the reviewers. Please see below; Fonts in blue have been provided for a point-by-point response to the reviewers' comments and concerns. Also, it should be noted that all of the editors' and reviewers' suggestions have been applied to the paper in addition to the reviewer's comments.
Comments and Suggestions for Authors
Reviewer #2:
The technical content of the paper is good enough but it should be improved a bit before publication. Some minor revisions are required
Authors’ Response: Thank you for your positive comment!
- Please remove acronyms from the abstract (ECC at least) and put it in the main body of the paper. In the abstract be as much general as possible (more an explanation of the research compared to a detailed technical description). In its present form, Abstract is very short. I suggest to put some more information regarding methodology and main results.
Authors' Response: Thank you for your constructive comment. (1) The researchers made an attempt to eliminate all acronyms from the abstract; nonetheless, they encountered a significant number of redundant terms, hence exacerbating the constraint of the abstract's word restriction. The authors made an effort to include all additional acronyms into the main body of the work in response to your insightful remarks. We kindly want your support and understanding.
(2) The abstract of the study has been revised by the authors in the following manner.
(Background) Shear damage of beams is typically brittle damage that is significantly more det-rimental than flexural damage. (Purpose) Based on the super high toughness and good crack control ability of Engineered Cementitious Composites (ECC), the shear performance of concrete-ECC beams is investigated by replacing a portion of the concrete in the tensile zone of reinforced concrete beams with ECC and employing high-strength reinforcing bars to design concrete-ECC beams and to clarify the shear performance of concrete-ECC beams. The purpose of this investi-gation is to elucidate the shear performance of concrete-ECC beams. (Methodology/approach) Experimental and finite element analyses were conducted on the shear performance of 36 web-less reinforced concrete-ECC composite beams with varied concrete strengths, shear-to-span ratios, ECC thicknesses, and interfacial treatments between the layers. (Results) The results indicate that the effect of the shear span ratio is greater, the effect of the form of interface treatment is smaller, the effect is weakened after the ECC thickness is greater than 70mm (i.e., the ratio of re-placement height to section height is approximately 0.35), the shear resistance is reduced when the hoop rate is greater, and the best shear resistance is obtained when the ECC 70mm thickness and the hoop rate of 0.29% are used together. (Conclusions) This study can serve as a technical reference for enhancing the problems of low durability and inadequate fracture control perfor-mance of RC beams in shear and as a guide for structural design research.
- Authors say “Shear resistance design has been a standard topic in structural design, and although the ductility characteristics of concrete can be increased by setting hoop reinforcement, the effect of transmitting shear force by the hoop and longitudinal reinforcement alone is less than ideal [5].” This is true. I suggest to consider also this very recent good reference DOI 10.1016/j.engstruct.2021.113483 that explains this topic in a deep way.
Authors’ Response: Thank you for pointing these out. The author adds this reference and revises it in the paper as follows:
Hippola et al. [6] proposed a novel finite element cell, which was then subjected to a comprehensive validation process including 170 tests to assess its correctness. The conducted tests exhibit variations in many key parameters, including shear span-to-depth ratios, rates of longitudinal and transverse reinforcement, concrete strengths, section depths, boundary conditions, and distinct mechanisms of damage. A novel finite element cell is developed with the aim of conducting a full investigation of the shear mechanism in reinforced concrete. Recent research has shown that the enhancement of existing reinforced concrete structures is not feasible without external interventions.
[6]Hippola H, Wijesundara K K, Nascimbene R. Response of shear critical reinforced concrete frames and walls under monotonic loading. Engineering Structures, 2022, 251: 113483.
- Authors say “In foreign countries” compared to what country ? The origin of the authors ? Please be more clear or change the sentence.
Response: We thank the reviewer for this questions..The author intended "abroad" to be a comparison with China. But this is an inaccurate statement. The author has modified this expression and added a more appropriate one.
- Please no need for colors in Table of a scientific paper. Please use eventually grey scale.
Authors’ Response: We thank the reviewer for bringing this to our attention. The author has changed all tables to grayscale.
- I suggest to improve quality of tables. Too big with a lot of text in different rows. Please check and modify accordingly.
Authors’ Response: Thank you for your constructive comment. The author has improved the quality of all tables.
It should be noted that this paper was reviewed by a total of three reviewers, and reviewer 1 suggested dividing Table 1 into three tables.

Reviewer 3 Report
This study focuses on an experiment to clarify shear performance of combined concrete-ECC beams without web reinforcement. The experiment was conducted with different concrete strengths, shear-to-span ratios, ECC thicknesses, and interlayer interface treatments as variables. Finally, the results of experiments are compared with that of finite element analysis by Abaqus software. The results are consistent within acceptable discrepancy. Overall, the damage characteristics of RC and ECC are similar. The variables of ECC composite beams except the interface treatment, shear-to-span ratio improved shear-related ability. This study advances our understanding of the effect of the variables on shear performance. However, the researchers should make the manuscript improved in terms of grammatical errors and some points considering the comments below.
Ÿ On page#1, line#36, it's unclear what "this damage mode" means, it needs to be described in detail.
Ÿ On page#1, line#42, there are no full spellings of “FRC”, “DRECC”, “SPECC”, etc. the author should provide the full spellings of the abbreviations when they appear first in the manuscript.
Ÿ On page#3, the images of Figure.1 are not aligned up and down, and some letters are cut off. The figures should be well-organized so that the potential readers can understand the manuscript better.
Ÿ On page#7, line#165, “are’ grammatically incorrect, it should be changed “is” because, “the concrete” is singular.
Ÿ On page#8, line#186, the verb “does” should be changed “do”.
Ÿ On page#12, the graphs of Figure 9 are not aligned up and down.
Ÿ On page #14 line#285, there is no “TableC.2.3-4”.
Ÿ On page #16, title on the vertical coordinate in the graphs of Figure 14 are in Chinese characters, they should be in English.
1)Is the work a significant contribution to the field?
The research is enough to understand ECC’s contribution for shear performance.
2)Is the work well organized and comprehensively described?
It can be understandable what the author intended but, descriptions should be in more detail.
3) Is the work scientifically sound and not misleading?
Overall, there are no misleading parts, but some need to be corrected.
4) Are there appropriate and adequate references to related and previous work?
The authors referred to appropriate papers on the variables applied to this experiment.
5) Is the English used correct and readable?
The English used in the paper is readable and understandable, but they could be much improved.
6) English language and style
The English in this paper is correct and meaningful.
Author Response
Response to Reviewers' Comments (Manuscript ID materials-2471297)
Experimental study of the shear performance of combined concrete-ECC beams without web reinforcement
Author's Response
Dear Editors and Reviewers,
Thank you for allowing us to submit a revised version of the manuscript "Experimental study of the shear performance of combined concrete-ECC beams without web reinforcement" for publication in the Scientific Reports. We sincerely appreciate the time and effort you and the reviewers dedicated to providing feedback on our manuscript and are grateful for our paper's insightful and constructive comments. The thorough review helped immensely in shaping and improvement of the manuscript. We have incorporated most of the suggestions made by the reviewers. Please see below; Fonts in blue have been provided for a point-by-point response to the reviewers' comments and concerns. Also, it should be noted that all of the editors' and reviewers' suggestions have been applied to the paper in addition to the reviewer's comments.
Reviewer #3
Reviewer comments
This study focuses on an experiment to clarify shear performance of combined concrete-ECC beams without web reinforcement. The experiment was conducted with different concrete strengths, shear-to-span ratios, ECC thicknesses, and interlayer interface treatments as variables. Finally, the results of experiments are compared with that of finite element analysis by Abaqus software. The results are consistent within acceptable discrepancy. Overall, the damage characteristics of RC and ECC are similar. The variables of ECC composite beams except the interface treatment, shear-to-span ratio improved shear-related ability. This study advances our understanding of the effect of the variables on shear performance. However, the researchers should make the manuscript improved in terms of grammatical errors and some points considering the comments below.
Authors' Response:
- On page#1, line#36, it's unclear what "this damage mode" means, it needs to be described in detail.
Authors’ Response: We thank the reviewer for these questions. The term "this damage mode" refers to the shear damage mode·.
- On page#1, line#42, there are no full spellings of “FRC”, “DRECC”, “SPECC”, etc. the author should provide the full spellings of the abbreviations when they appear first in the manuscript.
Authors’ Response: Thank you for pointing these out. Abbreviated names appearing for the first time in the manuscript have been changed to full spelling
- On page#3, the images of Figure.1 are not aligned up and down, and some letters are cut off. The figures should be well-organized so that the potential readers can understand the manuscript better.
Authors’ Response: Thank you for pointing these out. Revision of charts and graphs in the text has been completed.
- On page#7, line#165, “are’ grammatically incorrect, it should be changed “is” because, “the concrete” is singular.
Authors’ Response: Thank you for pointing these out. We have changed "are" to "is" on page 7, line 165.
- On page#8, line#186, the verb “does” should be changed “do”.
Authors’ Response: Thank you for pointing these out. We have changed the verb "does" to "do" on page 8, line 186.
- On page#12, the graphs of Figure 9 are not aligned up and down.
Authors’ Response: Thank you for pointing these out. We have aligned the top and bottom of the graph in Figure 9 on page 12.
- On page #14 line#285, there is no “TableC.2.3-4”.
Authors’ Response: Thank you for pointing these out. Completion has been added to the manuscript
- On page #16, title on the vertical coordinate in the graphs of Figure 14 are in Chinese characters, they should be in English.
Authors’ Response: Thank you for pointing these out. We have changed the title of the chart on page 16, Figure 14, in the vertical columns to English.
- Comments on the Quality of English Language
Authors’ Response: We appreciate this recommendation. The authors diligently evaluate and update the whole of the content, including aspects such as language, formatting, and punctuation, among others. This article aims to provide a comprehensive and intelligible analysis of the subject matter, with the intention of conveying valuable information to the reader.

Round 2
Reviewer 1 Report
Dear autors,
Thank you for take in count my suggests.
Reviewer 3 Report
All the comments made by reviewer have been well reflected in the revised manuscript.
The quality of the English language has been improved.